# High-Temperature and Drought Stress Effects on Growth, Yield and Nutritional Quality with Transpiration Response to Vapor Pressure Deficit in Lentil

**DOI:** 10.3390/plants11010095

**Published:** 2021-12-28

**Authors:** Noureddine El Haddad, Hasnae Choukri, Michel Edmond Ghanem, Abdelaziz Smouni, Rachid Mentag, Karthika Rajendran, Kamal Hejjaoui, Fouad Maalouf, Shiv Kumar

**Affiliations:** 1International Center for Agricultural Research in the Dry Areas (ICARDA), Rabat 10112, Morocco; hasnae_choukri@um5.ac.ma (H.C.); k.hejjaoui@cgiar.org (K.H.); 2Laboratoire de Biotechnologie et de Physiologie Végétales, Centre de Recherche BioBio, Faculté des Sciences, Mohammed V University Rabat, Rabat 10112, Morocco; abdelaziz.smouni@um5.ac.ma; 3AgroBioSciences (AgBS) Research Division, Mohammed VI Polytechnic University, Lot 660 Hay Moulay Rachid, Ben Guerir 43150, Morocco; michel.ghanem@um6p.ma; 4Biotechnology Research Unit, Regional Center of Agricultural Research of Rabat, National Institute of Agricultural Research (INRA), Rabat 10090, Morocco; rachidmentag@yahoo.ca; 5Vellore Institute of Technology (VIT), VIT School of Agricultural Innovations and Advanced Learning (VAIAL), Vellore 632014, Tamil Nadu, India; karthika.rajendran@vit.ac.in; 6International Center for Agricultural Research in the Dry Areas (ICARDA), Beirut 1108 2010, Lebanon; f.maalouf@cgiar.org

**Keywords:** lentil, drought stress, temperature stress, grain yield, limited transpiration trait, vapor pressure deficit, crude protein, biofortification, canopy temperature, zinc and iron

## Abstract

High temperature and water deficit are among the major limitations reducing lentil (*Lens culinaris* Medik.) yield in many growing regions. In addition, increasing atmospheric vapor pressure deficit (VPD) due to global warming causes a severe challenge by influencing the water balance of the plants, thus also affecting growth and yield. In the present study, we evaluated 20 lentil genotypes under field conditions and controlled environments with the following objectives: (i) to investigate the impact of temperature stress and combined temperature-drought stress on traits related to phenology, grain yield, nutritional quality, and canopy temperature under field conditions, and (ii) to examine the genotypic variability for limited transpiration (TR_lim_) trait in response to increased VPD under controlled conditions. The field experiment results revealed that high-temperature stress significantly affected all parameters compared to normal conditions. The protein content ranged from 23.4 to 31.9%, while the range of grain zinc and iron content varied from 33.1 to 64.4 and 62.3 to 99.3 mg kg^−1^, respectively, under normal conditions. The grain protein content, zinc and iron decreased significantly by 15, 14 and 15% under high-temperature stress, respectively. However, the impact was more severe under combined temperature-drought stress with a reduction of 53% in protein content, 18% in zinc and 20% in iron. Grain yield declined significantly by 43% in temperature stress and by 49% in the combined temperature-drought stress. The results from the controlled conditions showed a wide variation in TR among studied lentil genotypes. Nine genotypes displayed TR_lim_ at 2.76 to 3.51 kPa, with the genotypes ILL 7833 and ILL 7835 exhibiting the lowest breakpoint. Genotypes with low breakpoints had the ability to conserve water, allowing it to be used at later stages for increased yield. Our results identified promising genotypes including ILL 7835, ILL 7814 and ILL 4605 (Bakria) that could be of great interest in breeding for high yields, protein and micronutrient contents under high-temperature and drought stress. In addition, it was found that the TR_lim_ trait has the potential to select for increased lentil yields under field water-deficit environments.

## 1. Introduction

Lentil (*Lens culinaris* Medik.) is an important cool-season food legume that plays a significant role in human and animal nutrition as well as in maintaining and improving soil fertility [1,2]. Lentil grains are highly digestible and nutritious, representing a true staple crop in the dryland of many developing countries with affordable levels of dietary proteins (22–35%), vitamins, fiber, carbohydrates and essential micronutrients such as zinc and iron [3]. As lentil is highly nutritious and can be cooked more quickly than other pulses, it is the pulse most preferred by poor rural households worldwide, particularly those living in developing nations such as India, Bangladesh, Pakistan and Ethiopia [4]. However, micronutrient deficiencies are currently becoming more serious due to the high consumption of carbohydrate-rich cereal-based diets, which are low in micronutrients and vitamins [5,6]. Micronutrient malnutrition, commonly known as hidden hunger, affects more than 2 billion people in developing countries, while all over the world, >3 billion people suffer from micronutrient deficiencies [7,8,9,10,11]. In the last decade, efforts have been made to reduce the risks of malnutrition using several approaches including nutrient supplementation, diet diversification and food fortification. However, genetic biofortification using conventional and transgenic breeding methods has been recognized as the most effective and sustainable method for developing new crop varieties to combat hidden hunger and provide essential micronutrients and vitamins to the poor population through daily diet [6,12,13]. Highly nutritional genotypes have been developed through genetic biofortification in lentil [14], chickpea [15] and wheat [16], helping to minimize micronutrient deficiencies substantially in recent decades. However, the increase in the world population to around 10 billion by 2050 will further escalate malnutrition [17,18], and more efforts will be required to meet the human population’s nutritional needs for a healthy life. Enhancing the production of lentil would support food and nutritional security and improve the livelihoods of resource-poor farmers in developing nations. Still, lentil productivity in rainfed regions is expected to suffer from fluctuations in climate change, mainly due to the increased incidence of drought and higher temperatures [19].

High-temperature and drought stress have been reported as the major environmental factors that can markedly affect plant productivity and the quality of many cultivated crops [20]; however, their impact on cool-season food legumes appears to be more serious [21,22]. A temperature higher than 30 °C causes stress in most cool-season food legume crops. Both high-temperature and drought stress hamper plant growth by disturbing the normal physiology and morphology, thereby influencing an array of processes including growth, floral development, carbohydrates, protein content in grains, and micronutrient concentration (zinc and iron), which ultimately affect grain yield and nutritional quality [23,24]. Furthermore, the combined effect of high-temperature and drought stress on crops could be more severe than the individual stress impact. The reproductive stages of crops are more vulnerable to drought, high temperature and combined stress than the vegetative stages [14,25]. During anthesis and flowering periods, both stresses led to fertilization failures because of reduced pollen and ovule function and inhibited pollen development and sterility in legumes [26,27] and cereals [28,29]. While considering the current change in climate, it is expected that both the severity of high temperatures and the risk of drought will increase mainly in the subtropical region, impacting food security [30].

The impacts of high temperatures [31,32], water stress [33,34] and combined temperature-drought stress on lentil growth, yield and composition have been documented [20,25,35,36]. However, more information is needed on the physiological responses of lentil and their mechanisms to develop climate-resilient management strategies and new varieties. Since lentil is cultivated mainly in environments with high vapor pressure deficit (VPD) conditions (hot and dry areas) during the post-rainy season, where light rainfall is often noted, the study of the limited transpiration (TR_lim_) trait could be crucial in water-deficit conditions [37]. VPD is defined as the difference between the amount of moisture in the air and the moisture the air can hold when saturated. Temperature and relative humidity are the two major factors controlling VPD variation [38]. The high VPD, which usually occurs due to exacerbation of plant water stress in the middle of the day, affects plant photosynthesis, growth, yield and physiology [39]. The intensification of atmospheric water demand highly influences plant hydraulic status, and the key behind these impacts is plant stomatal behavior [40,41]. Regulation of stomatal conductance in high VPD conditions by limiting transpiration restricts the rate of water use and increases transpiration efficiency, helping to conserve water and support plant growth later in the season when drought develops [42,43]. TR_lim_ in response to high VPD has been reported in several crop species including wheat (*Triticum aestivum* L.) [41,44], maize (*Zea mays* L.) [45,46], sorghum (*sorghum bicolor* L.) [47,48], soybean (*Glycine max* L.) [49,50], peanut (*Arachis hypogaea* L.) [51,52], cowpea (*Vigna unguiculata* L.) [53,54], chickpea (*Cicer arietinum* L.) [55,56] and lentil [57]. Based on the available literature, the study by Guiguitant et al. [57] is the only investigation that examined lentil genotype differences in transpiration response to high VPD conditions. In the study by Guiguitant et al. [57], 17 lentil genotypes were tested for the expression of TR_lim_ and almost all tested lines showed a VPD breakpoint at approximately 3.4 kPa. In addition, Guiguitant et al. [57] reported a possible yield benefit of developing genotypes expressing the TR_lim_ trait, especially in regions with low rainfall, and that the impact of this trait on lentil productivity differs with geography and environments. Furthermore, several studies have shown increases in crop yield by 25 to 75% in water-limited areas based on crop simulation models [45,58,59,60]. Thus, developing and/or identifying drought- and heat-tolerant varieties is crucial in improving lentil yield under water-limited and high-temperature environments, which would directly contribute to food and nutritional security in the near future.

Based on the results from preliminary field screening [35], 20 lentil genotypes were selected as primary candidates for temperature and drought stress studies. Two independent experiments were conducted (i) to investigate the impact of high temperature and drought stress on traits associated with phenology, grain yield, nutritional quality and canopy temperature under field conditions, and (ii) to examine the possible genotypic variation in lentil for transpiration response to high VPD conditions over controlled environments. In the second experiment, transpiration was measured for plants subjected to different levels of VPD ranging from 1.20 to 4.50 kPa. The generated information may help to inform selection strategies and decisions within breeding programs and identify options to mitigate the impacts of high temperatures and water deficit on lentil productivity.

## 2. Results

### 2.1. Weather Data

Accumulated rainfall for normal planting was about 225 mm between December and April, while the late planting received around 26 mm mostly before the anthesis period. Under normal conditions, the minimum and maximum temperatures differed from −2.40 to 11.7 °C and 9.18 to 30.6 °C, respectively, during the vegetative stage, while they varied from 0.35 to 8.58 °C and 13.3 to 30.2 °C during the reproductive stage, respectively. Under the late planting, minimum and maximum temperatures were from 0.35 to 15.3 °C and from 13.3 to 37.5 °C respectively, before the flowering stage, and from 5.61 to 14.9 °C and from 22.6 to 36.9 °C, respectively, during the reproductive stage. Under normal planting, the RH ranged from 52 to 98% in the vegetative stage and 41 to 91% in the reproductive stage. It varied in late planting from 32 to 96% in the vegetative stage and from 42 to 79% during the reproductive stage. The maximum VPD registered in normal planting was 2.11 kPa and 3.25 kPa in late planting (Figure 1).

### 2.2. Combined Analysis of Variance

The combined analysis of variance indicated a significant difference among studied genotypes for all traits (Appendix A). ANOVA showed a highly significant (*p* < 0.001) effect of temperature stress and combined temperature-drought stress on all measured characteristics. However, genotype × treatment interaction was significant for all traits except NUPP and HSW. Furthermore, the coefficient of variation (CV) among treatments was less than 13% for most traits, but 35% for both GYP and NFPP, with 25 and 49% for NTPP and NUPP, respectively.

### 2.3. Temperature and Combined Heat-Drought Impact on Plant Phenology

PLH declined significantly, by 19.2%, in temperature stress and by 29.6% in the combined temperature-drought stress compared to its level in normal conditions. EGV ranged from 17.0 to 30.0 cm with an overall mean of 23.1 cm under normal conditions, whereas high-temperature and combined temperature-drought stress reduced EGV by 26.2 and 22.9%, respectively. DFF varied between 82 and 89 days in normal conditions, and a reduction of 28 days was observed for both stressed treatments. Likewise, days to 50% flowering (D50F) declined significantly, by 27 days, under both temperature and combined temperature-drought stress conditions (Table 1).

Temperature stress alone and in combination with drought severely reduced the D50P by 26 and 27 days, respectively. Under normal conditions, plants achieved physiological maturity in about 127 days, while days to maturity decreased significantly by 19 and 20 days under temperature stress and combined temperature-drought stress, respectively. NFPP varied from 40.8 to 234.2 in normal conditions and was significantly reduced by 44 to 69% in temperature stress and by 48 to 71% in combined temperature-drought stress, respectively. Similarly, NUPP decreased substantially by 21–62% and 3–43% in temperature stress and combined temperature-drought stress, respectively. NTPP ranged from 56 to 240 pods in normal conditions with an average of 122.8, while it decreased significantly by 44–72% under temperature stress and 46–75% under combined temperature-drought stress conditions. GYP ranged from 1.59 to 8.86 g with an overall mean of 4.13 g under normal conditions. The late planting treatments showed a significant reduction in grain yield, by 43% under temperature stress and 49% under the combined stress. HSW varied from 1.93 and 4.97 g in normal conditions, with a mean of 2.64 g, whereas it was decreased significantly by 8% and 19% under the temperature stress and combined temperature-drought stress, respectively (Table 1).

### 2.4. Influence of High-Temperature and Combined Temperature-Drought Stress on Fe, Zn and Protein Contents

Crude protein (CP) varied from 23.4 to 31.9% under normal conditions, with a mean of 28.9%. CP decreased significantly to 15% due to temperature stress and 53% under combined temperature-drought stress. Under normal conditions, the mean Zn concentration was 47.9 mg kg^−1^, ranging from 33.1 to 64.4 mg kg^−1^. High-temperature stress and combined temperature-drought stress significantly reduced Zn concentration by 14 and 18%, respectively. Similar results were observed for Fe with a reduction of 15% due to temperature stress and 20% under combined temperature-drought stress. Fe concentration varied from 62.3 to 99.3 mg kg^−1^ under normal conditions (Table 1).

### 2.5. Correlation Coefficient between Measured Traits

A significant positive correlation was observed between PLH and EGV (r = 0.38) under normal conditions. However, a highly significant negative correlation (*p* < 0.01) was observed among EGV, DFF and D50P. HSW had a negative correlation with DFF, D50F and D50P, and it had a positive correlation with EGV (r = 0.62, *p* < 0.01). GYP was positively correlated with NFPP and NTPP (r = 0.97 and r = 0.96 at *p* < 0.01, respectively). For quality traits, there were no correlations between CP and phenological and yield parameters under normal conditions. Positive correlations were observed between Fe and Zn (r = 0.68), D50P (r = 0.52) and DM (r = 0.38, *p* < 0.05), and negative correlations with EGV (r = −0.61, *p* < 0.01), HSW (r = −0.39, *p* < 0.05) and CP (r = −0.35, *p* < 0.05). In addition, a highly negative correlation was observed between Zn and CP (r = −0.47, *p* < 0.01) in normal conditions (Appendix A).

A significant negative correlation was found between EGV and DFF, D50F, and D50P under both temperature stress and combined temperature-drought stress (Appendix A). Similarly to normal conditions, GYP had a significant positive correlation with NTPP and NFPP (r = 0.90) under the two stressed conditions. Under temperature stress, Fe content showed a positive correlation with D50F (r = 0.37, *p* < 0.05), GYP (r = 0.34, *p* < 0.05) and Zn (r = 0.46, *p* < 0.01). A significant and positive correlation between Zn and Fe (r = 48, *p* < 0.01) was also observed in combined temperature-drought stress conditions. In addition, Zn was found to be positively correlated with HSW (r = 0.34, *p* < 0.05) and negatively correlated with CP under temperature stress (r = −0.36, *p* < 0.05) and combined temperature-drought stress (r = −0.45, *p* < 0.01). Likewise, a positive correlation was observed between CP and DM (r = 0.36, *p* < 0.05) under temperature stress.

### 2.6. Principal Component Analysis for Stress and Normal Conditions

Principal component analysis (PCA) was performed to study the relationships between traits in the studied lentil genotypes. Under normal conditions, the first three PCAs (PC1: 31.3%, PC2: 22.6% and PC3: 16.5%) explained 70.4% of the total variability (Table 2). The results of PCA showed that EGV, D50P, HSW and Fe content were the most important traits contributing to PC1 of distinct origin. In PC2, which described 22.6% of the total variance, NFPP, GYP and D50F demonstrated large contributions, while Zn content and CP accounted for much of the total variance in PC3 (Appendix A). These 20 genotypes were clustered into three groups based on hierarchical cluster analysis (Appendix A). The check Bakria (ILL 4605) and three genotypes (ILL 5919, ILL 7813 and ILL 3484) were identified in group 1 and had the highest GYP (5.55 g) and HSW (3.27 g), and moderate Zn, Fe and CP. In group 2, ten genotypes were classified and recorded the highest Fe (85.1 mg kg^−1^) and Zn (54.1 mg kg^−1^), with medium GYP and HSW. Six genotypes were grouped in cluster 3 and exhibited the highest CP (29.9%), medium Fe (72.4 mg kg^−1^) and HSW (2.5 g), and the lowest GYP (Table 3). The distribution of genotypes and measured traits along the first PC axes under normal conditions are shown in Figure 2.

For temperature stress, PC1 and PC2 (PC1: 34.0% and PC2: 24.3%) explained 58.3% of the total variation, while PC3 showed 16.5% of the total variation. PC1 was positively correlated with DFF, D50F, D50P, HSW and Zn, whereas PC2 was positively correlated with GYP, NTPP, NFPP and Fe content. Furthermore, negative correlations were observed between PC1 and EGV, NTPP and NFPP. PC3 had a significant positive correlation with Zn content and negative association with CP and DM (Table 2). Hierarchical cluster analysis was conducted, and the genotypes were grouped into three clusters (Appendix A). Cluster 1 grouped only the check Bakria (ILL 4605), a moderate heat- and drought-tolerant variety, which had the highest HSW (4.95 g) and Zn and Fe contents with a mean of 56.6 and 73.8 mg kg^−1^, respectively. Five heat-tolerant genotypes (ILL 7814, ILL 6104, ILL 7835, ILL 7833 and ILL 6338) and one moderately heat- and drought-tolerant genotype (ILL 6363) were identified in cluster 2. This group had the highest GYP with a mean of 3.30 g and CP with an average of 24.7%, while Zn and Fe concentrations were 42.5 and 69.7 mg kg^−1^, respectively. Thirteen genotypes including two heat-tolerant lines (ILL 8029 and ILL 7286), three combined temperature-drought tolerant genotypes (ILL 6362, ILL 7804 and ILL 6075), five moderately tolerant lines and three susceptible lines were found in cluster 3 (Appendix A). This group had moderate GYP with an average of 1.97 g, HSW with 2.35 g and CP with a mean of 24.4%, whereas Zn and Fe concentrations were 39.3 and 64.8 mg kg^−1^, respectively (Table 3). The biplot of PC1 and PC2 of the three clusters and traits under temperature stress conditions is shown in Figure 3.

The first three PCAs in the combined temperature-drought stress explained 66.2% of the entire dissimilarity (PC1:34.5%, PC2:19.3% and PC3:12.4%). PC1 showed a significant positive correlation with D50F, D50P, DM, GYP, NTPP, NFPP and CP, and a negative correlation with Zn, Fe and EGV. PC2 was positively correlated with GYP, NTPP and NFPP, and negatively correlated with D50F and HSW (Table 2). Based on hierarchical cluster analysis (Appendix A), 11 genotypes including two heat-tolerant lines (ILL 6104 and ILL 8029), four heat-drought tolerant lines (ILL 6075, ILL 7804, ILL 7835 and ILL 7814), three moderately tolerant lines (ILL 7223, ILL 7819 and ILL 8025) and three susceptible genotypes (ILL 5919, ILL 7813 and ILL 7820) were found in group 1. This group was characterized by the highest Zn and Fe concentrations with an average of 41.3 and 65.0 mg kg^−1^, respectively, and medium GYP (1.84 g). Group 2 comprised two moderately tolerant genotypes (ILL 4605 and ILL 6359), and had the highest CP and HSW, with a mean of 14.2% and 3.56 g, respectively. Three heat-tolerant lines (ILL 7833, ILL 7286 and ILL 6338), two moderately tolerant (ILL 6363 and ILL 3484) and one heat-drought tolerant genotype (ILL 6362) were grouped in cluster 3. This group had the highest GYP with an average of 2.97 g, and moderate concentrations of Zn and Fe with a mean of 37.1 and 59.9 mg kg^−1^, respectively (Table 3). The biplot of PC1 and PC2 of the three clusters and measured traits under the combined temperature-drought stress is illustrated in Figure 4.

### 2.7. Canopy Temperature Variation under Normal and Stress Conditions

Analysis of variance for canopy temperature (CT) showed a significant difference between genotypes and treatments; however, there were no significant differences for genotype x treatment interaction (Appendix A). Under normal conditions, the CT of plants increased significantly from an average of 29.1 °C at day 100 to 31.8 °C at day 115 after sowing. Under normal conditions, the minimum and maximum CT were 26.6–32.7 °C at day 100 and 29.7–33.4 °C at day 115 (Appendix A). Under temperature stress, the plants recorded an average of 29.2 °C and 31.7 °C at day 65 and day 70, respectively. The average CT increased significantly from 31.5 °C at day 80 to 31.9 °C at day 90. The lowest maximum CT was recorded at day 65 with 31.3 °C, while the highest was achieved at day 90 with 33.8 °C. Under combined temperature-drought stress, the CT increased significantly from an average of 29.5 °C at day 65 to 31.9 at day 70. On day 80 and day 90, the CT was 31.6 and 31.7 °C, respectively. Plants at day 65 recorded the lowest maximum CT (31.3°C), while, the highest was recorded at day 90 with 34.0 °C.

### 2.8. Transpiration Response to VPD under Controlled Environments

The results revealed that nine lentil genotypes were well represented by the two-segment regression with a breakpoint (BP) (Table 4). However, 11 genotypes exhibited a linear response and failed to express the limited transpiration (TR_lim_) trait during the increase in VPD (Table 5). The R^2^ of the genotypes with BP ranged from 0.60 to 0.87 and varied from 0.66 to 0.95 for the genotypes without TR_lim_. The results displayed high variation between the assessed genotypes in TR to increasing VPD. The value of BP ± SE of the genotypes expressing TR_lim_ varied from 2.76 ± 0.43 to 3.51 ± 0.54 kPa with an average of 3.14 ± 0.66 (Table 4).

High dissimilarity was distinguished for the slopes below and above BP increasing TR with further increases in VPD. The slope at VPD above BP ranged from −1.63 ± 5.34 to 28.1 ± 11.3 mg with an average of 14 mg H_2_O m^−2^ s^−1^ kPa^−1^. In contrast, the slope below BP varied from 18.1 ± 5.49 to 44.7 ± 7.37 mg with an average of 34.5 mg H_2_O m^−2^ s^−1^ kPa^−1^. Genotype ILL 8029 showed the highest BP at about 3.51 kPa, while the two genotypes ILL 7835 and ILL 7833 recorded the lowest BP at around 2.76 and 2.81 kPa, respectively (Figure 5). The genotype ILL 7833 expressed the lowest TR during the increase in VPD, with 18.1 and −1.63 mg H_2_O m^−2^ s^−1^ kPa^−1^ for left and right slopes, respectively. Similarly, ILL 7814 and ILL 7835 were characterized by a very low TR after expressing the BP, with slopes greater than the BP ranging from 5.59 to 10.8 mg H_2_O m^−2^ s^−1^ kPa^−1^. Among genotypes that did not express a BP, the check Bakria (ILL 4605) had the lowest slope with a TR of 16.5 ± 1.51 mg H_2_O m^−2^ s^−1^ kPa^−1^. Oppositely, ILL 8025 exhibited the highest slope with 36.5 ± 2.78 mg H_2_O m^−2^ s^−1^ kPa^−1^. The results also revealed that ILL 5919, ILL 7813 and ILL 8025 had the maximum TR at around 175 mg H_2_O m^−2^ s^−1^ kPa^−1^, which lost TR_lim_ trait expression during the increase in VPD. ILL 6262 recorded the highest TR with 210 mg H_2_O m^−2^ s^−1^ kPa^−1^, followed by the genotypes ILL 6338 and ILL 3484, which had a TR of 190 mg H_2_O m^−2^ s^−1^ kPa^−1^ (Table 4).

## 3. Discussion

### 3.1. Phenology, Yield and Nutritional Quality under High-Temperature and Drought Stress

In this study, delayed sowing to expose the reproductive stage to heat stress significantly affected phenological traits, grain yield components, nutritional quality parameters and canopy temperature in lentil genotypes. However, combining drought with heat stress through limited irrigation had a more severe impact on plant growth, resulting in significantly lower grain yield and nutritional quality. These results are consistent with our previous studies that confirmed the influence of water limitation and high temperature on lentil [14,35]. Similar findings were also observed in chickpea [61], lentil [31,36], soybean [62] and groundnut [63] where combined temperature-drought stress was more detrimental than temperature stress alone.

Our findings revealed that the number of filled pods decreased because of high-temperature and combined temperature-drought stress. This is in agreement with earlier studies that showed temperature and/or drought stress negatively influenced the reproductive processes, mainly pollen and ovule fertility [64,65], resulting in the reduction of filled pods. Recently, Jiang et al. [66] described that temperature stress reduced the number of pollen grains per anther, induced smaller pollen grains and increased reactive oxygen species (ROS) production in pollen grains in pea. Still, it did not affect ROS accumulation in ovules and ovule number per ovary. Furthermore, high-temperature exposure when young floral buds were visible at the first formed reproductive node was more destructive to flower retention, seed set, pod development and seed yield compared to heat exposure started later, when flowers at the second reproductive node were fully open. Temperature and drought stress also impact pollen vacuolization, which plays a vital role in increasing the volume of pollen grains with the accumulation of cytoplasmic components [67]. A recent study by Fábián et al. [29] showed that the combination of high temperature and water stress altered the phenology of the plants, reduced pollen viability, shortened the duration of gametogenesis and grain filling, and modified the morphology and anatomy of the pistils. Functionality of female and male reproductive parts was reduced by 34 and 66%, respectively.

Our study showed a decrease in seed weight due to temperature stress, while combined temperature-drought stress magnified the impact. It is a fact that during the seed-filling stage, the decline in photosynthesis and leaf sucrose metabolism in response to temperature stress and drought stress led to lower carbohydrate availability for import into developing seeds [68]. Previous investigations suggested seed weight is reduced in response to high-temperature and/or drought stress during grain filling by altering endosperm cells [69] and reducing starch accumulation [70]. Starch is the central component in seeds for major global staple crops. A decrease in source strength and carbon availability for starch biosynthesis through the grain filling period will definitely reduce seed size and weight [71,72]. Recent studies revealed a decline in the activity of enzymes involved in starch biosynthesis under high-temperature or water stress. Eventually, their combination contributed to a decrease in the starch accumulation rate and duration and starch content [73,74]. Sehgal et al. [36] indicated a significant decrease in starch content under temperature stress and water stress in lentil seeds. However, the combined temperature-drought stress caused the most severe reduction due to the inhibition of starch synthesizing enzyme (starch synthase) and sucrose synthesizing activity (sucrose synthase), which were previously identified to have a correlated effect on seed size in chickpea [75].

A significant reduction in protein content was reported under temperature stress; however, the combined temperature-drought stress caused more decrease, which is in agreement with previous observations in lentil [14,36], and in chickpea [76] through reduced function of PSII, weakened nitrogen anabolism and strengthened protein catabolism. Previous findings by Zahedi et al. [77] and Triboï et al. [78] revealed a stable relation between protein content and the total quantity of nitrogen in wheat grain. They suggested that the synthesis of storage proteins is limited mainly by the availability of nitrogen. Similarly, Liu et al. [79] also reported a decrease in crude protein fractions in drought-stressed alfalfa (*Medicago sativa* L.) due to water limitation in addition to a reduction in N fixation. For that, improvement of N fixation could lead to higher biomass production and water-use efficiency and may increase pod yield under drought stress conditions.

High temperature induced a significant reduction in grain micronutrient concentrations. The combined temperature-drought stress further aggravated the decline, by 18% for Zn and 20% for Fe, compared to that under normal conditions. Similar findings were observed in other legume crops such as lentil, where iron and zinc contents were dramatically reduced in response to temperature and drought stress due to decreased root nutrient uptake, with reduced root biomass and metabolic rate [80] or by direct damage to roots [81]. In addition, the reduction in transpiration rate because of water deficit may also decrease nutrient absorption and the effectiveness of their use by the plants [36,82]. Choukri et al. [14] suggested that the decrease in iron and zinc was attributed to decreasing water availability under heat and drought stress conditions. Furthermore, Hummel et al. [83] reported that the variations in zinc, iron and protein under drought stress conditions are more affected by weather conditions than genotype. The present study also revealed a negative correlation between Zn and protein content in high-temperature stress and heat-drought stress conditions, and Fe and protein content in the combined temperature-drought stress, which are in disagreement with previous studies conducted by Ghanbari et al. [84] and Impa et al. [85]. They reported a positive correlation between protein content and micronutrients under temperature and drought stress conditions. However, Fe and Zn were positively correlated under the three treatments, which was also reported in earlier findings in lentil [86], chickpea [87,88] and pea [89]. Furthermore, our results showed a significant increase in canopy temperature of plants under high-temperature stress and combined temperature-drought stress. This effect might be explained by inhibition of stomatal conductance and transpiration reducing leaf water content, which resulted from an increase in leaf temperatures [90,91].

### 3.2. Transpiration Response to High VPD under Controlled Conditions

Our findings showed significant genetic variation among lentil genotypes in transpiration response to increasing VPD under controlled environments. Eleven genotypes consistently increased TR with increasing VPD, while nine exhibited a distinct response by limiting their TR when VPD reached about 3.1 kPa. Our outcomes indicated that most of the genotypes displayed lower BP compared to what was reported by Guiguitant et al. [57], who showed a BP at approximately 3.4 kPa in the majority of lentil genotypes, with the lowest BP at 3.31 kPa. In our study, two tolerant genotypes (ILL 7833 and ILL 7835) limited their TR at early VPD (2.8 kPa), representing the lowest BP over all the already published VPD experiments in lentil. Restriction of TR under high VPD was also reported in many other species such as chickpea [56], peanut [51], soybean [49], sorghum [92] and maize [93], in which the BP values mostly fluctuated from 1.1 to 2.7 kPa. However, Schoppach and Sadok [94] showed high BP ranging from 2.4 to 3.9 kPa in wheat, while Sinclair et al. [95] described a VPD threshold ranging from 2.6 to 3.38 kPa in peanut.

Over the whole range of tested VPD, almost all of the tolerant genotypes showed lower TR_lim_ compared to the sensitive lentil lines, which exhibited a continuously increasing TR with increasing VPD. These results support our field findings [35], and they are in good agreement with the conclusion of Belko et al. [54] in cowpea. Genotypes with TR_lim_ may have considerable potential for increased soil water conservation, indicating somehow more effective control of TR under limited-water conditions [43]. Several findings reported that genotypes with a conservative water behavior have the opportunity of using the conserved soil water to preserve physiological activity during the grain filling period and produce higher grain yield than genotypes without the TR_lim_ trait in late-season water deficit conditions [37,42,43,57,60,96].

Interestingly, the check variety Bakria, a moderately tolerant line to drought and heat stress, exhibited the lowest TR among all tested lines with 16 mg H_2_O m^−2^ s^−1^ kPa^−1^ under increasing VPD, and probably had the lowest conductance among the studied genotypes. Among the tolerant lines, ILL 7833 and ILL 8029 had the lower TR presented in slope 1 with 18.1 and 22.7 H_2_O m^−2^ s^−1^ kPa^−1^, respectively. However, genotype ILL 8029 registered the maximum BP at 3.51 kPa, which was likely to be disadvantageous in expressing the TR_lim_ trait under high levels of VPD [94,97]. Variation in TR shown in the two slopes, which normalized water use efficiency, can be explained by the difference in the hydraulic conductance to water flux in the plants. Several studies have indicated that the TR_lim_ trait under high evaporative conditions of VPD is likely due to low hydraulic conductivity in leaves between the xylem and into the guard cells, which increased water use efficiency [93,98,99]. The data described here do not allow support for the claim of a hydraulic conductance limitation in leaves; however, the difference in VPD breakpoint values among lentil genotypes may indicate variation in stomatal conductance between studied genotypes.

Since the experiment was conducted under controlled conditions, the variations in temperatures and relative humidity were the main key factors to reach higher VPD levels; however, the high temperatures (>35 °C) may influence TR response of the plants. A recent study by Kar et al. [100] revealed that an increase in VPD had a powerful positive impact on plant water loss irrespective of the effect of temperature, light and relative humidity. Although previous studies have reported similar observations, the effects of temperature, relative humidity, radiation and water on TR had minor effects compared to VPD [94,101,102].

Genotypes with TR_lim_ under high VPD offer an essential candidate for breeding programs to improve lentil yields in water deficit areas [57]. A low BP represents an imperative approach by the plants to maintain the greatest water conservation during critical drought periods when high VPD promotes TR. Our outcomes indicate that studied lentil genotypes could be used to improve yield potential over a range of drought-stressed environments. Hence, tolerant genotypes such as ILL 7833 and ILL 7835, which had the lowest BP, may be desirable for drought-prone conditions and improve lentil productivity in regions affected by drought stress. The TR profiles generated from the current investigation present an opportunity to understand the physiological behaviors of tolerant and sensitive lentil genotypes under controlled conditions. However, additional studies are needed to examine more physiological and genetic responses of lentil material under field and controlled conditions.

## 4. Materials and Methods

### 4.1. Plant Material

The material consisted of 20 lentil genotypes selected from our previous field screening study [35], where we evaluated a focused identification of germplasm strategy (FIGS) set against drought and temperature stresses. In the present study, we included moderately heat tolerant, moderately drought tolerant, heat tolerant, heat sensitive, drought tolerant and drought sensitive lines (Appendix A).

### 4.2. Field Experiment

The germplasm was assessed under three independent experiments at the ICARDA experimental station Marchouch (33.56° N, 6.63° W, 392 m altitude) during the 2016–17 cropping season. These three experiments were deemed to represent three treatments, namely (i) normal date of planting (Treatment A), (ii) late planting with irrigation (Treatment B) and (iii) late planting without irrigation (Treatment C). All of the treatments were planted in an alpha lattice design with two replications. In the three treatments, each genotype was planted in a three-row plot of 1 m length, with a spacing of 0.30 m between rows. In each row, seeds were sown at 2 cm depth, maintaining 10 cm space between plants. Treatment A resulted in optimal growing conditions (150 mm well-distributed rainfall and temperature below 27 °C, without any heat or water stress to the plants). Treatment B (planted 65 days after normal planting date with irrigation at field capacity throughout crop duration) imposed high-temperature stress. In both late planting treatments, the plants were synchronized with temperatures above 32 °C during the reproductive stage. Regular irrigation at field capacity avoided any water stress to the plants during their growth and development. Treatment C (planted 65 days after normal planting date without irrigation during the reproductive stage) imposed combined high-temperature and drought stress. Treatment A was planted on 27 December 2016, without any supplementary irrigation during the cropping period, as the crop received enough well-distributed rainfall during the cropping season. Treatments B and C were planted on 1 March 2017. Irrigation was performed regularly to sustain water supply at field capacity using a sprinkler system throughout the crop duration in treatment B. In contrast, irrigation was stopped at the flower initiation stage onward in treatment C to impose water stress (<5 mm rainfall during the reproductive stage) combined with the temperature stress. All field management followed standard agricultural practices in lentil production [103].

### 4.3. Data Collection

Data were collected on early growth vigor (EGV) after 1 month of sowing, plant height (PLH), days to first flowering (DFF), days to 50% of flowering (D50F), days to 50% of podding (D50P) and days to physiological maturity (DM) on a plot basis. Five plants were randomly selected from each plot to measure number of total pods plant^−1^ (NTPP), number of filled pods plant^−1^ (NFPP) and number of unfilled pods plant^−1^ (NUPP), grain yield plant^−1^ (GYP) and hundred-seed weight (HSW). Canopy temperature (CT) was determined using a thermal infrared camera FLIR T4xx-series (Model T420-KIT-15). Thermal images were captured between 11:00 and 13:00 GMT time on a sunny day and analyzed using FLIR tools+ software (FLIR systems, Teledyne FLIR, Wilsonville, OR, USA). The seeds harvested from these experiments were analyzed for protein content and micronutrient contents (zinc and iron).

### 4.4. Crude Protein Content

A 0.3 g amount of ground lentil seeds from each treatment was digested with sulfuric acid and selenium mixture following the modified Kjeldahl procedure [104]. Based on the color reaction between ammonium and a weakly alkaline mixture of sodium salicylate and sodium hypochlorite, a UV visible spectrophotometer was used to determine the color development in the samples at 560 nm. Next, crude protein content (CP) was measured using nitrogen values multiplied by 6.25, and triplicate analyses were performed for each sample.

### 4.5. Iron and Zinc Determination

Seeds were ground by a Cyclone mill (Twister, 10 mm–250 um, Retsch). Iron (Fe) and zinc (Zn) concentrations were measured using a modified HNO_3_ and H_2_O_2_ method [105]. In the digestion block (QBlock series, Horiba), 0.5 g of each ground sample was placed in individual tubes and digested with 6 mL nitric acid (HNO_3_), followed by heat treatments at 90 °C for 1 h. To each tube, 3 mL of 30% hydrogen peroxide (H_2_O_2_) was added, and sample digestion was continued by heating for 15 min at 90 °C, and then 3 mL of 6 M hydrochloric acid (HCL) was added. Once samples were cooled, the solutions were filtered and diluted to 10 mL with distilled water. The mineral content analysis was carried out by inductivity coupled plasma-optical emission spectrometry (ICP-OES); (iCAP-7000 Duo, Thermo Fisher Scientific, Waltham, MA, USA) at the Cereals and Legumes Quality Laboratory, ICARDA, Rabat, Morocco.

### 4.6. Plant Growth Conditions in the Greenhouse

An independent experiment was conducted during summer 2018 in the greenhouse at ICARDA, Rabat, Morocco. Three seeds were sown at a depth of 2.5 cm in plastic pots (15 cm in diameter and 20 cm in height) filled with 1.5 kg of 50% sandy loam soil and 50% of compost garden soil (Floragard Vertriebs-GmbH product, Oldenburg) that included 18-10-20 N-P-K fertilizer. Here, the same set of 20 genotypes assessed under field conditions were studied under controlled conditions. Five replications were used for each genotype. The pots were positioned in a randomized design on steel grid platform installed at mid-height of 1.3 m. In each pot, a small hole was opened at the bottom of the end cap to facilitate drainage. Plants were grown under well-watered conditions, and the temperature in the greenhouse was maintained at 25 °C/18 °C (day/night) for 1 month. The photosynthetic photon flux density in the greenhouse was approximately 600 μmol m^−2^ s^−1^.

The day before the measurement, pots were watered to full capacity until their water flowed from the bottom of the pots, then the pots were allowed to drain overnight. After drainage, the pots were bagged with plastic bags and covered with polyethylene balls to keep soil evaporation to a minimum. A thermo-hygrograph sensor (Tinytag Ultra 2 TGU-4500 Gemini Datalogger Ltd., Chichester, UK) was positioned over the plant to measure the temperature and RH at 5 min intervals. Hourly atmospheric VPD was calculated on the basis of average temperature and RH. Plants were exposed to each humidity and temperature treatment for 1 h and then reweighed to measure the final weight. On an hourly basis, the transpiration rate (TR) was calculated as the weight difference between successive measurements using a balance with a resolution of 0.1 g. The measurement was conducted in the morning from 7:00 am (Morocco standard time) at low VPD until 7:00 pm, when the VPD decreased following the midday maximum. Measurements were first started with the low VPD (0–1.5 kPa) treatment, the medium VPD treatment (1.5–2.5 kPa), and finally, the high VPD (2.5–4.5 kPa) treatment. The experiment in the greenhouse allowed us to control the temperature and RH, exposing the plants to the expected ranges of VPD (1.18 to 4.5 kPa). The average temperature and RH during the experiment fluctuated from 19 to 40 °C and from 30 to 80%, respectively. A high VPD level was achieved by increasing temperature and decreasing RH inside the greenhouse. At the end of the experiment, the leaf area was measured using a leaf area meter, and the TR was expressed as water loss per unit of leaf area and time (g H_2_O_2_ cm^−2^ h^−1^).

### 4.7. Statistical Analysis

#### 4.7.1. Field Data Analysis

Analysis of variance was carried out using the general linear model (GLM) in IBM SPSS statistics 23. Duncan’s post-hoc was applied to compare differences between the mean values at *p* < 0.05. Pearson’s correlation coefficient was determined for the three treatments. Principal component analysis (PCA) was performed using the *Factoextra* and *FactoMineR* packages in R version 4.1.0 and RStudio version 1. 3.1093 [106,107]. In addition, hierarchical cluster analysis was performed using Ward’s squared Euclidean distance method with the *dendextend* R package [108].

#### 4.7.2. Vapor Pressure Deficit Analysis

The individual data for each VPD treatment of each genotype were used in the regression analysis of transpiration response. The data were first tested to fit the data to a two-segment linear regression using GraphPad Prism 7 (GraphPad Software Inc.). The results of a successful regression fit to the two-segment model were the coefficients defining two intersecting linear regressions:If VPD < BP, TR = intercept 1 + slope 1 (VPD)(1)
If VPD ≥ BP, TR = intercept 2 + slope 2 (VPD)(2)
where BP is the breakpoint between the two linear segments; it is also an output of the GraphPad analysis and an estimate of the standard error for BP. The intercept represents the constant of the first and second line segments; the two slopes were statistically compared within GraphPad for a significant difference at *p* ≤ 0.05 level. In case there was a significant difference, the two-segment model characterized the outcomes for that genotype. All data for a genotype were assumed to fit a single linear regression model if two slopes were not identified to be significantly different.

## 5. Conclusions

The outcomes of the present study validate our previous field screening results. Almost all tolerant genotypes showed a high adaptation to high temperatures and combined temperature-drought stress under field conditions and exhibited a fairly clear TR_lim_ at high VPD under controlled conditions. These genotypes would contribute to water saving in soil and improve the productivity under terminal heat or water-limited environments. Interestingly, two tolerant genotypes (ILL 7833 and ILL 7835) exhibited the lowest breakpoint found in lentil, which may be exclusively desirable for harsh regions affected by water deficit. Genotypes such as ILL 7835, ILL 7814 and Bakria (ILL 4605) combine good nutritional quality with moderate to high yield under both heat and combined heat-drought stress, which is of great interest for breeding programs. The parameters of the tested lentil genotypes in response to high temperatures, water deficit and higher VPD would facilitate formulating new experimental strategies to shed light on the molecular and genetic basis of heat and drought tolerance mechanisms in the future. Hence, combining crop physiology with molecular techniques could be a promising approach to improve lentils under high temperatures, water deficit and higher VPD.

## Figures and Tables

**Figure 1 plants-11-00095-f001:**
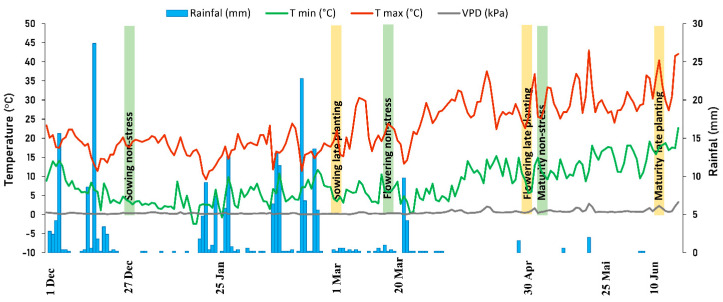
Rainfall, maximum and minimum temperatures, and vapor pressure deficit (VPD) during normal and late planting experiments in Marchouch, Morocco during 2016/17 season.

**Figure 2 plants-11-00095-f002:**
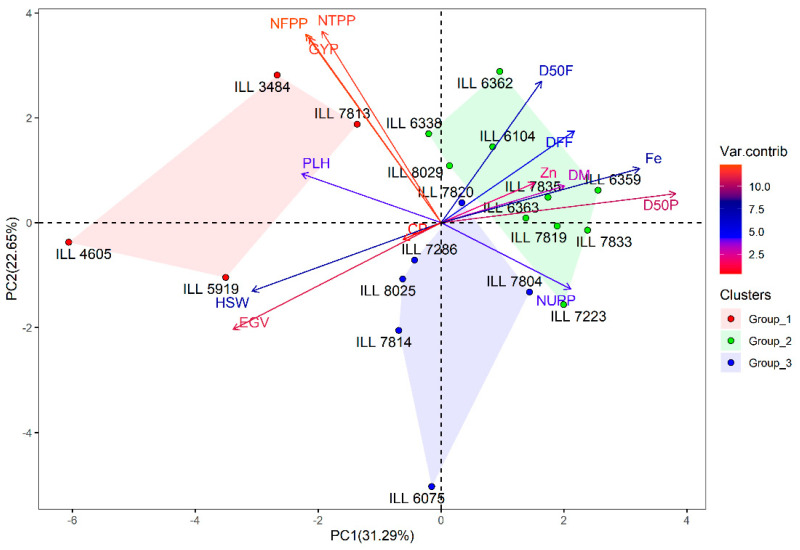
Biplot of the first two axes of the PCA analysis summarizing the relationship between the measured variables and the classification of lentil genotypes under normal conditions.

**Figure 3 plants-11-00095-f003:**
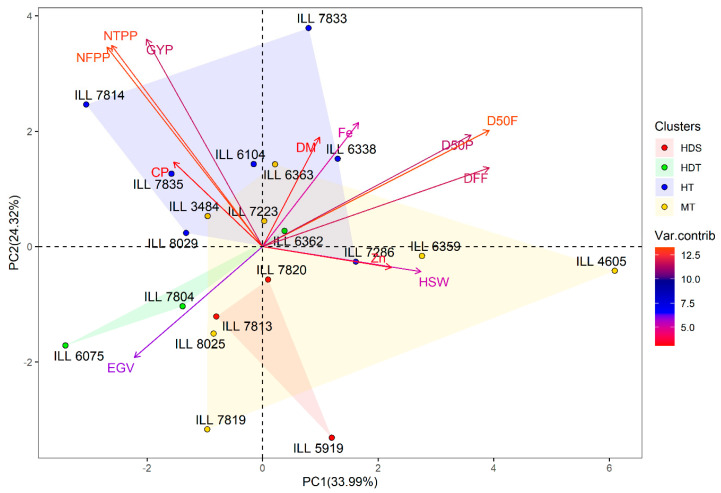
Biplot of the first two axes of the PCA analysis summarizing the relationship between the measured variables and the classification of lentil genotypes under temperature stress conditions. HDS, heat stress and combined heat-drought susceptible; HDT, heat-drought tolerant; HT; heat tolerant and MT, moderately tolerant to heat stress and combined heat-drought stress.

**Figure 4 plants-11-00095-f004:**
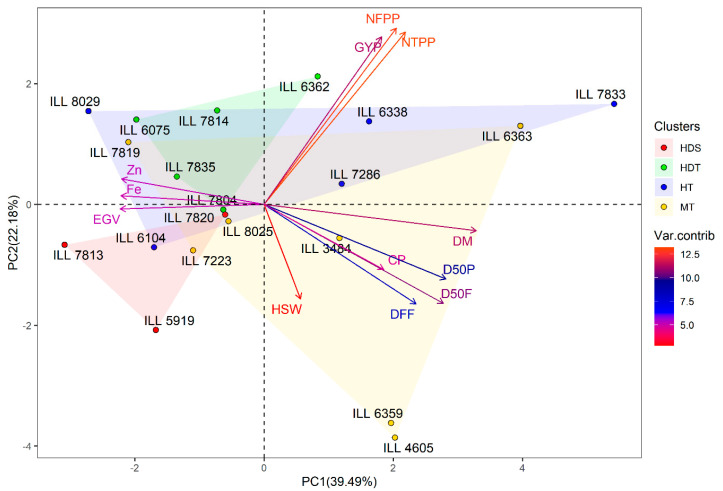
Biplot of the first two axes of the PCA analysis summarizing the relationship between the measured variables and the classification of lentil genotypes under the combined temperature-drought stress conditions. HDS, heat-drought susceptible; HDT, heat-drought tolerant; HT; heat tolerant and MT, moderately tolerant to heat-drought stress.

**Figure 5 plants-11-00095-f005:**
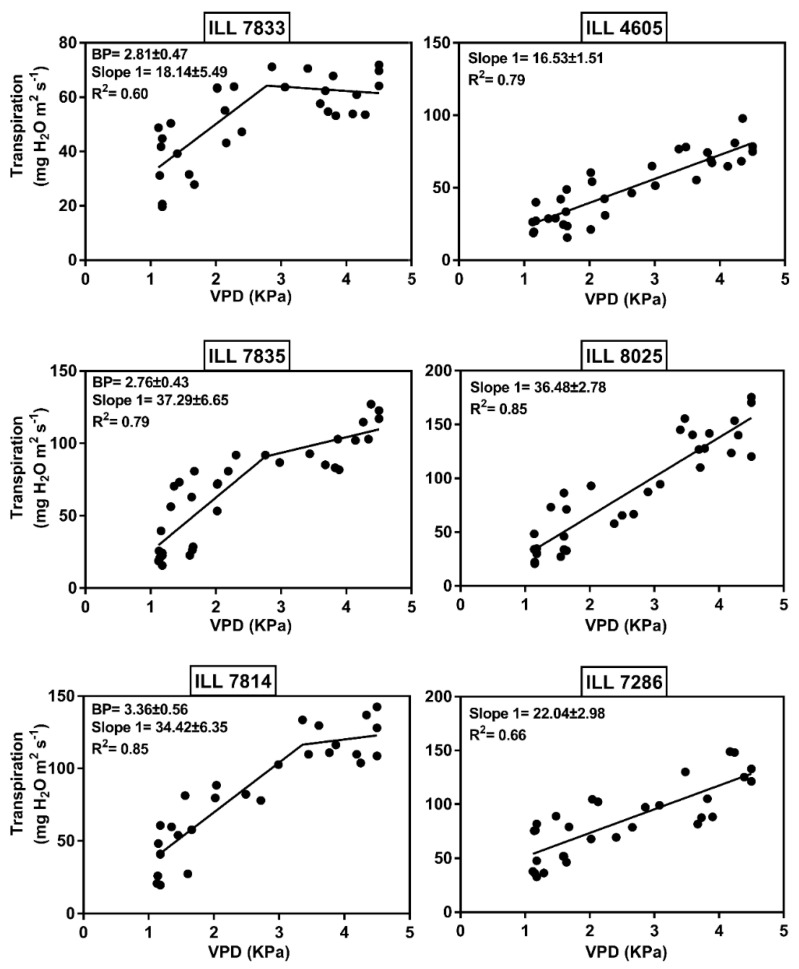
Transpiration rate (mg H_2_O m^−2^ s^−1^) response to increasing vapor pressure deficit (VPD in kPa). Genotypes ILL 7833, ILL 7835 and ILL 7814 represent tolerant accessions with two segmental regression, while ILL 4605, ILL 8025 and ILL 7286 represent susceptible genotypes with linear transpiration response to increasing VPD. The breakpoint (BP), slopes and R^2^ are indicated.

**Table 1 plants-11-00095-t001:** Minimum, maximum and mean morphological parameters of 20 lentil genotypes under normal, temperature stress and temperature-drought stress conditions.

Trait	Normal			Temperature Stress			Temperature-Drought Stress		
	Min	Max	Mean ± SE	Min	Max	Mean ± SE	Min	Max	Mean ± SE
PLH	24.75	34.21	28.28 ^a^ ± 3.39	19.75	27.25	22.86 ^b^ ± 0.30	14.33	25.03	19.74 ^c^ ± 0.37
EGV	17.00	30.00	23.12 ^a^ ± 0.56	13.00	21.00	17.07 ^b^ ± 0.30	12.00	21.00	17.82 ^c^ ± 0.35
DFF	82.00	89.00	84.95 ^a^ ± 0.22	60.00	66.00	61.55 ^b^ ± 0.19	590.00	64.00	61.15 ^b^ ± 0.23
D50F	86.00	100.00	93.08 ^a^ ± 0.45	65.00	74.00	68.53 ^b^ ± 0.31	630.00	74.00	68.32 ^b^ ± 0.39
D50P	98.00	106.00	101.78 ^a^ ± 0.36	71.00	80.00	76.20 ^b^ ± 0.44	700.00	80.00	74.77 ^c^ ± 0.41
DM	124.00	131.00	127.20 ^a^ ± 0.32	101.00	105.00	103.53 ^b^ ± 0.17	100.00	105.00	101.70 ^c^ ± 0.17
NFPP	40.83	234.25	111.45 ^a^ ± 7.43	12.75	129.40	57.75 ^b^ ± 4.66	12.00	119.50	52.07 ^b^ ± 4.14
NUPP	3.50	25.25	11.34 ^a^ ± 0.88	1.33	20.00	6.15 ^b^ ± 0.51	2.00	24.50	7.89 ^b^ ± 0.79
NTPP	56.00	240.00	122.79 ^a^ ± 7.23	15.75	134.00	63.90 ^b^ ± 4.71	14.00	130.00	59.96 ^b^ ± 4.17
GYP	1.59	8.86	4.13 ^a^ ± 0.28	0.43	4.56	2.35 ^b^ ± 0.17	0.66	4.32	2.12 ^b^ ± 0.14
HSW	1.93	4.97	2.64 ^a^ ± 0.09	1.59	5.23	2.42 ^b^ ± 0.11	1.42	5.51	2.15 ^c^ ± 0.13
CP	23.40	31.90	28.91 ^a^ ± 0.35	22.00	27.00	24.36 ^b^ ± 0.20	11.00	14.90	13.34 ^c^ ± 0.16
Zn	33.10	64.40	47.86 ^a^ ± 1.52	27.40	59.10	41.12 ^b^ ± 1.41	30.40	50.90	39.28 ^c^ ± 0.80
Fe	62.30	99.30	78.19 ^a^ ± 1.44	51.30	81.60	66.71 ^b^ ± 1.42	52.40	73.50	62.86 ^c^ ± 0.81

Means for each variable followed by the same letters are not significantly different (*p* < 0.05). PLH, plant height; EGV, early growth vigor; DFF, days to first flowering; D50F, days to 50% flowering; D50P, days to 50% podding, DM, days to 95% maturity; NFPP, number of filled pods plant^−1^; NUPP, number of unfilled pods plant^−1^; NTPP, number of total pods plant^−1^; GYP, grain yield plant^−1^; HSW, hundred-seed weight; CP, crude protein; Zn, zinc content and Fe, iron content.

**Table 2 plants-11-00095-t002:** Eigenvalues and eigenvectors of the three PCA dimensions under normal conditions, temperature stress and the combined temperature-drought stress.

Traits	Normal			Temperature Stress			Temperature-Drought Stress		
	PC1	PC2	PC3	PC1	PC2	PC3	PC1	PC2	PC3
PLH	−0.52 *	0.21 ns	−0.07 ns	−0.31 ns	−0.02 ns	0.01 ns	−0.27 ns	−0.13 ns	0.19 ns
EGV	−0.78 **	−0.47 *	0.05 ns	−0.49 *	−0.42 ns	0.38 ns	−0.64 **	−0.05 ns	0.14 ns
DFF	0.50 *	0.40 ns	−0.50 *	0.86 **	0.30 ns	0.02 ns	0.64 **	−0.44 ns	0.18 ns
D50F	0.38 ns	0.62 **	−0.58 **	0.86 **	0.44 ns	0.01 ns	0.74 **	−0.44 *	0.44 ns
D50P	0.88 **	0.13 ns	0.06 ns	0.79 **	0.42 ns	−0.21 ns	0.77 **	−0.33 ns	0.30 ns
DM	0.46 *	0.16 ns	0.45 *	0.22 ns	0.41 ns	−0.70 **	0.89 **	−0.11 ns	0.19 ns
NFPP	−0.50 *	0.83 **	0.14 ns	−0.59 **	0.76 **	0.23 ns	0.55 *	0.81 **	0.16 ns
NTPP	−0.44 *	0.84 **	0.13 ns	−0.57 **	0.76 **	0.23 ns	0.59 **	0.78 **	0.09 ns
NUPP	0.48 *	−0.29 **	−0.08 ns	0.05 ns	0.20 ns	0.04 ns	0.22 ns	−0.20 ns	−0.13 ns
GYP	−0.49 *	0.81 **	0.11 ns	−0.44 ns	0.79 **	0.33 ns	0.49 *	0.76 **	0.19 ns
HSW	−0.71 **	−0.30 ns	0.42 ns	0.60 **	−0.09 ns	0.42 ns	0.13 ns	−0.50 *	0.35 ns
CP	−0.14 ns	−0.07 ns	−0.65 **	−0.34 ns	0.32 ns	−0.60 **	0.52 *	−0.30 ns	−0.39 ns
Zn	0.35 ns	0.18 ns	0.77 **	0.49 *	−0.08 ns	0.61 **	−0.61 **	0.14 ns	0.35 ns
Fe	0.74 *	0.24 ns	0.51 **	0.36 ns	0.47 *	0.44 ns	−0.64 **	0.05 ns	0.55 *
Eigenvalue	4.38	3.17	2.31	4.08	2.92	1.98	4.83	2.70	1.73
Variance explained (%)	31.29	22.65	16.49	34.00	24.32	16.52	34.51	19.30	12.42
Total variance (%)	31.29	53.29	70.43	34.00	58.32	74.85	34.51	53.81	66.23

** 0.01, * 0.05, ns not significant. PLH, plant height; EGV, early growth vigor; DFF, days to first flowering; D50F, days to 50% flowering; D50P, days to 50% podding, DM, days to 95% maturity; NFPP, number of filled pods plant^−1^; NUPP, number of unfilled pods plant^−1^; NTPP, number of total pods plant^−1^; GYP, grain yield per plant; HSW, hundred-seed weight; CP, crude protein; Zn, zinc content; Fe, iron content.

**Table 3 plants-11-00095-t003:** Grain yield, hundred-seed weight, crude protein, zinc content and iron content of the three clusters for normal, temperature stress and combined temperature-drought stress.

Cluster	Experiment	GYP	HSW	CP	Zn	Fe
Cluster I	Normal	5.55	3.27	29.78	44.12	69.50
	High-temperature	1.65	4.95	22.52	56.63	73.85
	Temperature-drought	1.84	2.00	12.92	41.26	65.02
Cluster II	Normal	4.06	2.44	27.94	54.13	85.15
	High-temperature	3.30	2.17	24.66	42.49	69.73
	Temperature-drought	1.28	3.56	14.25	33.85	58.65
Cluster III	Normal	3.30	2.54	29.94	39.94	72.39
	High-temperature	1.97	2.35	24.37	39.30	64.77
	Temperature-drought	2.97	1.96	13.87	37.12	59.99

GYP; grain yield, HSW; 100-seed weight, CP; crude protein, Zn; zinc content and Fe; iron content.

**Table 4 plants-11-00095-t004:** Outputs from the analysis of the two-segment linear regression used for fitting the TR to the VPD variations.

Genotype	Breakpoint		Left Slope		Right Slope		R^2^
	BP ± SE	CI (95%)	S1	CI (95%)	S2	CI (95%)	
ILL 3484	3.27 ± 0.95	3.01 to 4.50	44.48	32.68 to 59.19	24.95	3.45 to 39.94	0.87
ILL 6075	3.15 ± 1.14	2.73 to 4.51	28.91	21.74 to 45.84	17.37	0.25 to 28.46	0.82
ILL 6104	3.00 ± 0.25	3.00 to 3.45	36.40	25.72 to 47.06	12.26	−2.57 to 26.43	0.83
ILL 6338	3.37 ± 0.61	3.00 to 3.45	43.73	31.41 to 60.63	15.42	−9.64 to 38.61	0.81
ILL 6362	3.01 ± 0.99	2.76 to 3.46	44.71	34.40 to 71.39	28.14	2.88 to 41.59	0.87
ILL 7814	3.36 ± 0.56	2.79 to 4.17	34.42	25.90 to 77.81	5.59	−26.00 to 28.45	0.85
ILL 7833	2.81 ± 0.47	1.67 to 3.57	18.14	9.59 to 38.87	−1.63	−13.49 to 8.32	0.60
ILL 7835	2.76 ± 0.43	2.23 to 3.30	37.32	23.76 to 50.88	10.79	−3.32 to 24.91	0.79
ILL 8029	3.51 ± 0.54	3.07 to 4.01	22.69	14.10 to 31.29	13.10	−10.92 to 37.13	0.70

The BP is the breakpoint (kPa), S1 and S2 are the left slope and right slope, respectively (mg H_2_O m^−2^ s^−1^ kPa^−1^), CI is the confidence interval.

**Table 5 plants-11-00095-t005:** Results of the 11 genotypes exhibiting single linear regression fits of TR response to vapor pressure deficit. The table includes slope with standard error (SE), X-intercept and R^2^ values.

Genotype	Slope ± SE	Slope CI (95%)	X-Intercept	X-Intercept CI (95%)	R^2^
ILL 5919	27.23 ± 2.59	21.96 to 32.50	−0.58	−1.34 to −0.06	0.77
ILL 4605	16.53 ± 1.51	13.46 to 19.61	−0.39	−1.13 to 0.13	0.79
ILL 6359	31.18 ± 2.82	25.44 to 36.92	−0.13	−0.82 to 0.36	0.79
ILL 6363	27.07 ± 2.72	21.55 to 32.58	−0.75	−1.65 to −0.15	0.74
ILL 7223	25.95 ± 1.01	23.88 to 28.02	0.06	−0.20 to 0.28	0.95
ILL 7286	22.04 ± 2.98	15.94 to 28.13	−1.32	−2.84 to −0.44	0.66
ILL 7804	30.32 ± 2.17	25.93 to 34.71	0.12	−0.35 to 0.49	0.82
ILL 7813	29.08 ± 3.67	21.54 to 36.62	−0.15	−1.19 to 0.48	0.70
ILL 7819	24.22 ± 2.27	19.62 to 28.83	−0.20	−0.92 to 0.30	0.76
ILL 7820	23.01 ± 2.61	17.60 to 28.43	−0.61	−1.62 to 0.03	0.78
ILL 8025	36.48 ± 2.78	30.81 to 42.16	0.22	−0.27 to 0.59	0.85

## Data Availability

Data is contained within the article and Appendix A.

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
