# Peer review of "High-Temperature and Drought Stress Effects on Growth, Yield and Nutritional Quality with Transpiration Response to Vapor Pressure Deficit in Lentil"

_plants, 2021, doi:10.3390/plants11010095_

Round 1
Reviewer 1 Report
The present manuscript present some useful results regarding effect of heat and drought stress yield and nutritional quality with transpiration response to vapor pressure deficit under controlled conditions in lentil, However, the manuscript need revision particularly in method section (see comments annotated in attached file)

Reviewer 2 Report
Very well concieved manuscript and presented in a very best possible way.
Here are some minor mistakes: Line 20- must be among major limitations Line 29- Remove was before ranged Line 31 replace was with ranged Line 72 replace through with though Furthermore the manuscript looked okAuthor Response
Please find attached file

Reviewer 3 Report
The manuscript is very interesting and contains a lot of valuable information. The presented study consist of original research results. It describes the effect of high temperature and drought stress on traits associated with phenology, seed yield, nutritional quality and canopy temperature under field conditions, and investigates possible genotypic variation in lentil for transpiration response to high vapor pressure deficit conditions over controlled environments.
The title accurately reflect the content of the article, but is too long. I suggest removing "under controlled conditions". This information is in the “Material and Methods” section.
I also suggest to improve the keywords on: lentil; vapor pressure deficit; heat-drought stress; high-temperature stress; transpiration; protein content; micronutrients
The research problem and the purpose of the work have been formulated correctly. The ‘Introduction’ part is good, although the statement that there is the only investigation of Guiguitant et al. that examine lentil genotype differences in transpiration response to high VPD conditions seems risky. Maybe it is better to write: "in the available literature ...". In addition, the item number from the reference list must be placed next to the name Guiguitant et al. (line: 113-114, 115).
The ‘Results’ section is presented in a clear manner and correctly interpreted and the statistical analyzes are carried out correctly, but I found some issues that should be addressed.
Table 1 – the abbreviations in the table (NFPP, NUPP) do not agree with the explanation below the table.
Table 2 – Instead of Dim.1, Dim.2 etc, please insert PCA1, PCA2 etc, according to the descriptions in the manuscript.
Line 235 i 237 - Is Fe the highest in cluster 2 and 3? Is the GYP the lowest in cluster 3, not medium?
Line 259, 288, 449 – (Bakria ILL4605) - there are only numbers on the Fig. 2-5, so please include them.
Line 290 – ILL6368 is correct? There is no such line in Fig. 4
Line 342-344 – “ILL 6262 recorded the highest TR with 210 mg 342 H2O m-2 s-1 kPa-1, followed by the genotypes ILL 6338 and ILL 3484 which had a TR of 190 343 mg H2O m-2 s-1 kPa-1 (Table 5).”
There are no such genotypes in Table 5.
Besides, temperature can be entered with 1 decimal place.
‘Discussion’ and ‘Conclusion’ sections are correct.
The 'Material and methods' section is sufficiently detailed, but there is no information on how the thermal stress plants was applied to the plants during flowering under field conditions (control 27oC, stress 32oC)?
The references are properly selected.
